# Mapping the physiological changes in sleep regulation across infancy and young childhood

**Lachlan Webb** [1,2]*, **Andrew J. K. Phillips**[3], **James A. Roberts**[1,2]

**1** Brain Modelling Group, QIMR Berghofer Medical Research Institute, Herston, Brisbane, Queensland, Australia, **2** Faculty of Medicine, University of Queensland, Queensland, Australia, **3** Flinders Health and Medical Research Institute (Sleep Health), Flinders University, Bedford Park, South Australia, Australia

* Lachlan.Webb@qimrberghofer.edu.au

**Data Availability Statement:** Data availability Sleep/wake patterns in Infants 1-4 are available at https://github.com/brain-modelling-group/infant-sleep-trajectory. All four datasets are in public domain: Infant 1 (Github repository https://github.

## Abstract

Sleep patterns in infancy and early childhood vary greatly and change rapidly during development. In adults, sleep patterns are regulated by interactions between neuronal populations in the brainstem and hypothalamus, driven by the circadian and sleep homeostatic processes. However, the neurophysiological mechanisms underlying the sleep patterns and their variations across infancy and early childhood are poorly understood. We investigated whether a well-established mathematical model for sleep regulation in adults can model infant sleep characteristics and explain the physiological basis for developmental changes. By fitting longitudinal sleep data spanning 2 to 540 days after birth, we inferred parameter trajectories across age. We found that the developmental changes in sleep patterns are consistent with a faster accumulation and faster clearance of sleep homeostatic pressure in infancy and a weaker circadian rhythm in early infancy. We also find greater sensitivity to phase-delaying effects of light in infancy and early childhood. These findings reveal fundamental mechanisms that regulate sleep in infancy and early childhood. Given the critical role of sleep in healthy neurodevelopment, this framework could be used to pinpoint pathophysiological mechanisms and identify ways to improve sleep quality in early life.

## Author summary

Sleep is crucial for healthy neurodevelopment in infants and young children. Sleep patterns in these early years vary greatly between infants, and can also change rapidly with age. While the underlying mechanisms of adult sleep are well explored, it is not well understood how those mechanisms change during infancy and early childhood. We used a well-established mathematical model of sleep regulation to explore how maturing subcortical sleep circuitry could explain the changes in sleep patterns in the longitudinal sleep/wake data of four infants. Our results identified changes in the circadian rhythm's influence on sleep, the dynamics of sleep pressure accumulation, and a delayed response of the circadian rhythm to light being required to produce similar sleep patterns to those

com/jiuguangw/Agenoria, commit be11a8e on 30/12/2021, used for academic purposes under CC BY-NC-SA 4.0), Infant 2 (Github repository https://github.com/jitney86/Baby-data-viz, commit d92043f on 11/8/2017, used for academic purposes under MIT license), Infant 3 (Infant #01 in Figure 1 of [24], doi: 10.1016/j.infbeh.2005.11.001), and Infant 4 (Figure 3 of [25], doi: 10.1093/sleep/22.3.303). Code availability MATLAB code for the model is accessible on GitHub at https://github.com/brain-modelling-group/infant-sleep-trajectory.

**Funding:** J.A.R. was supported by the National Health and Medical Research Council (Project Grants 1145168 and 1144936). The funders had no role in study design, data collection and analysis, decision to publish, or preparation of the manuscript.

**Competing interests:** I have read the journal's policy and the authors of this manuscript have the following competing interests: L.W. and J.A.R. declare no competing interests. A.J.K.P. has received research funding from Versalux and Delos, he is an inventor on two patents related to control of sleep-wake patterns, and he is co-founder and co-director of Circadian Health Innovations PTY LTD.

of the real-life infants. This modelling is a step towards a better understanding of infant and young child sleep.

## Introduction

Sleep is fundamental to health in all stages of life and critical for neurodevelopment in early life. Across many species, sleep patterns change dramatically from early life to adulthood, likely reflecting differential functions of sleep and changing brain morphology across the lifespan [1,2]. Sleep in human infancy is characterized by very long sleep durations (>12 h) and the division of sleep into multiple bouts per day. As an infant ages, sleep bouts begin to consolidate, total sleep time decreases [3], and more sleep occurs during the night [4]. Night-time sleep consolidates during the first few years of childhood, with fewer awakenings, and day-time sleep bouts (naps) ultimately cease for most (though not all) children by age 5 years [5]. This results in a single, consolidated block of night-time sleep, which persists through adolescence into adulthood. Surprisingly, it is still unknown which biological mechanisms drive these striking changes in sleep patterns.

The neural circuitry that regulates sleep/wake patterns has been well characterized in mammals [6]. Neuronal populations in the brainstem and hypothalamus serve as an 'ascending arousal system' that promotes wakefulness via neuromodulation of the corticothalamic system. Mutual inhibition between these wake-promoting neurons and distinct populations of sleep-promoting neurons gives rise to a 'sleep/wake switch', which drives the distinction between sleep and wake states [6–8]. Within this sleep/wake switch framework, sleep/wake patterns are generally assumed to be driven by a combination of the circadian process (originating from the brain's central clock, the suprachiasmatic nuclei; SCN) [8,9] and the sleep-homeostatic process (driven by accumulation of sleep-promoting factors in the brain such as adenosine) [8]. Both of these processes are known to directly act upon the sleep/wake switch [6]. Changes in sleep patterns across early life could therefore be driven by changes to the circadian process, the sleep homeostatic process, or the circuitry of the sleep/wake switch itself.

Mathematical models have long been used to understand the mechanisms underpinning sleep patterns. Even before the physiology of the sleep/wake switch was elucidated, it was known that circadian and sleep homeostatic processes regulate sleep patterns in human adults [10], and that differences in dynamics of either the circadian or sleep homeostatic process could cause changes in sleep duration and the number of sleep bouts per day [11]. More recently, physiological models of the sleep/wake switch have been developed [12–14], providing a powerful framework for linking sleep phenotypes (both healthy and pathological) with underlying physiological mechanisms. This approach has provided insight into the basis for individual differences in sleep and circadian timing in both healthy and disease states [15–18], narcolepsy [19], sleep fragmentation [20], and differences in sleep patterns between species [21,22]. Furthermore, these models have provided insights into the physiological processes that drive age-related changes in sleep patterns from adolescence to old age [16,23]. To date, however, we know very little about the physiological mechanisms that cause the profound changes to sleep patterns during neurodevelopment in infants and children.

Here, we used a physiologically based model of the sleep/wake switch to identify the mechanisms by which sleep patterns change across development from infancy through childhood. We posited that parameters of the homeostatic and circadian drives in particular—for example as implicated in adolescent development age [23]—are able to explain the major changes in sleep patterns across development. We hence fitted the model to longitudinal sleep data across

infancy to determine the trajectories of physiological parameters that underpin sleep maturation.

## Results

### Sleep/Wake switch model

We used a previously validated model of the sleep/wake switch that includes the sleep homeostatic process and a dynamic model of the circadian clock ([16]; see Methods for more details). This is a mean-field model describing the ensemble activity of populations of neurons, and produces sleep/wake behaviour through a mutual inhibition between sleep-promoting neurons (the ventrolateral preoptic group; VLPO) and wake-promoting neurons (monoaminergic group; MA) of the ascending arousal system. This switch is acted upon by the sleep homeostatic process and the central circadian clock in the SCN, which is in turn affected by light.

### Empirical sleep patterns

We examined longitudinal densely-sampled sleep activity data from four public domain datasets (Fig 1), each for a single infant:

1. Sleep diary data from 2 days to 548 days post birth (approx. 18 months);

2. Sleep diary data from 86 days to 535 days post birth (approx. age 3–18 months).

3. Actigraphy data from 7 days to 372 days post birth (approx. 12 months) [24].

4. Sleep diary data from 3 days to 182 days post birth (approx. 6 months) [25].

   Details about the data sources and processing are in the Methods.

   The common features of these data are: (1) erratic and not well entrained sleep patterns immediately after birth, (2) eventual consistent sleep patterns containing daytime naps at similar times each day, and (3) changes in sleep patterns as the number of naps decreases and the infant transitions into a new sleep pattern with different sleep timings.

### The parameter space of sleep maturation

One of the primary defining characteristics of infant sleep is the increased amount of sleep compared to adults (approximately 13–15 h per day during first few months of life), a characteristic that typically decreases with development to around 13 h per day at the age of 2 years, to 11–12 h throughout childhood, and 7–9 h in adolescence [26,27]. Moreover, immediately after birth, sleep is broken into a large number of bouts [4], before consolidating to fewer bouts of longer length. As an initial model exploration, we calculated total sleep duration (TSD) and number of sleep bouts per 24 h day (BPD) as a function of four model parameters via a systematic grid search. The four key parameters we chose to vary were based on previous evidence that these parameters may underlie individual differences in sleep timing and structure [21,23,28,29]: $\mu$, the rate of increase of sleep homeostatic pressure, which when increased will increase TSD and BPD; $\chi$, the characteristic clearance time for sleep homeostatic pressure, which when decreased will increase BPD; $v_{vc}$, the strength of the circadian drive (inhibition to the VLPO), which when increased in magnitude (more negative) reduces BPD and advances/delays sleep/wake patterns with respect to the daily light curve; and $b$, the phase-delay bias of the circadian clock to light, which when increased delays sleep/wake patterns. A selection of results from this grid search is presented in Fig A in S1 Text. One plausible trajectory through the space that captures the main features of sleep development consists of: an initial high $\mu$ that

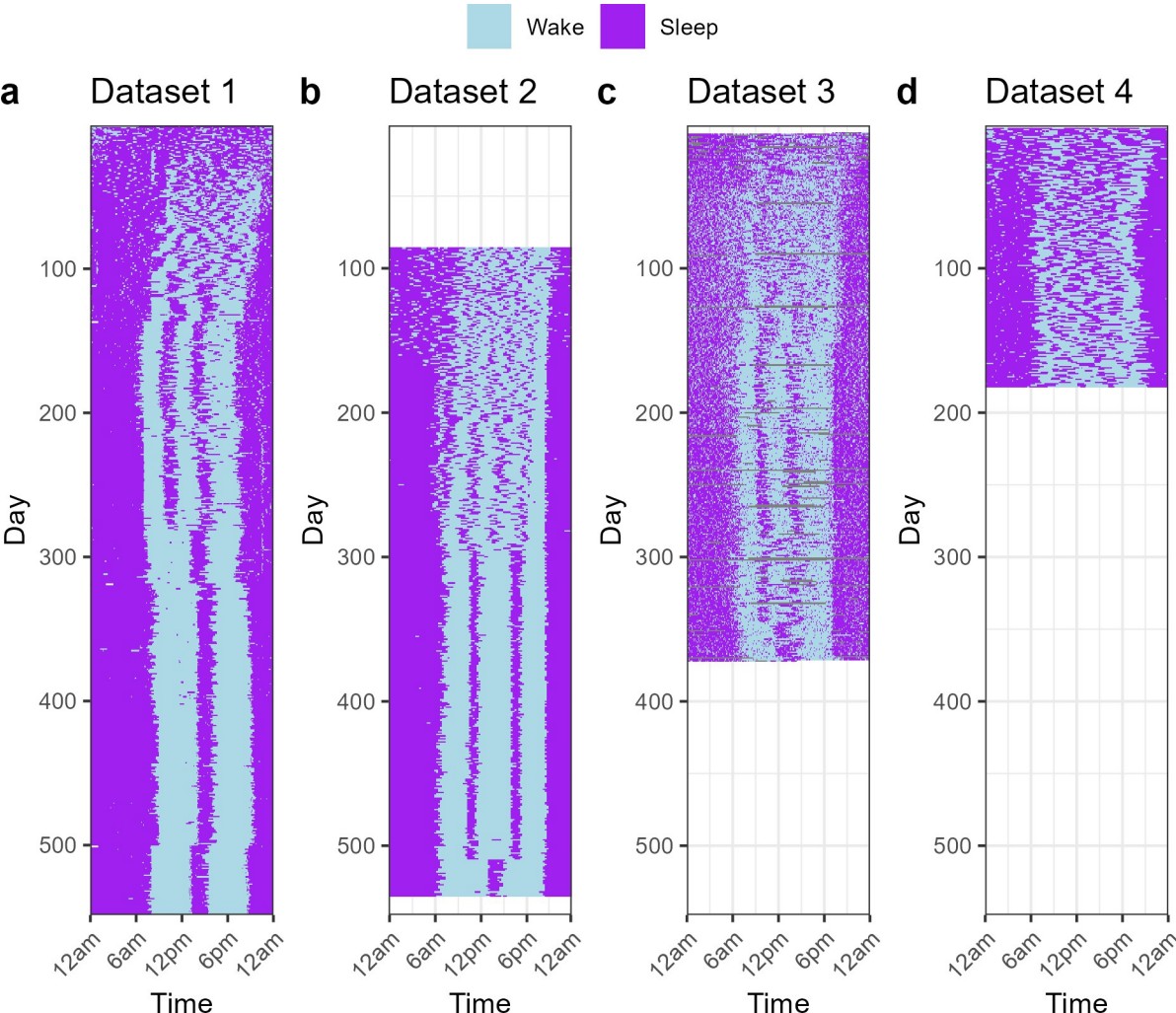

**Fig 1. Empirical sleep/wake data from four infants.** *a)* Sleep diary data from the Agenoria GitHub repository. *b)* Sleep diary data from the Baby-data-viz GitHub repository. *c)* Sleep patterns derived from actigraphy data from Jenni, Deboer and Achermann [24]. *d)* Sleep diary data from McGraw, Hoffmann, Harker and Herman [25]. See Methods for further details.

decreases with age (but does not reach the adult value of $\sim 4.2$ nM s during childhood), a $\chi$ that starts low but rapidly increases, and increases in the magnitude of $v_{vc}$ (more negative).

Skeldon, Derks and Dijk [23] has previously found that sleep patterns in late childhood (age 10+) corresponded with a higher $\mu$. Changes in $v_{vc}$ have already been suggested in the literature [30–32], and in particular McGraw, Hoffmann, Harker and Herman [25] and Jenni, Deboer and Achermann [24] showed that the circadian influence on sleep tends to be close to zero in the days soon after birth as the infant loses the mother's circadian signal and their own circadian system (already active in utero [33,34]) matures in response to the ex utero environment.

## Trajectories through parameter space inferred from empirical data

We next sought trajectories in model parameters that reproduce the specific empirical sleep patterns of Fig 1. Using the output from the previously mentioned grid search, we sought to find parameter combinations that minimized the difference between modelled and empirical

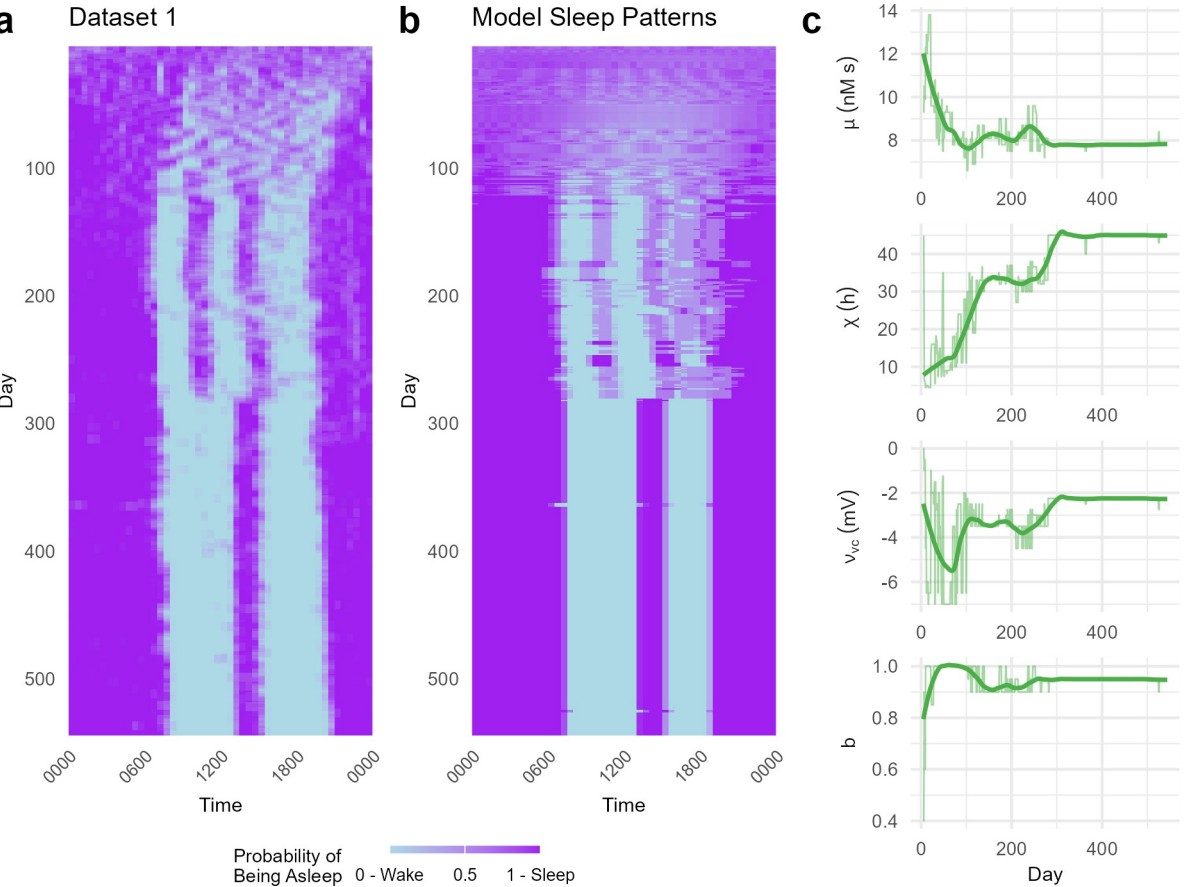

**Fig 2. Fitted trajectory of sleep maturation. *a)*** The sleep/wake patterns of Infant 1 summarised by the probability of being asleep per 30-minute window, averaged across 7 days with 6 day overlap (7 day sliding window). ***b)*** The sleep/wake patterns for each best-fitting parameter combination for each set of sleep probabilities in a). ***c)*** The trajectories of best-fitting parameter combinations, superimposed with a Loess smoothed line (thick lines) with smoothing parameter $\alpha_{Loess} = 0.2$.

probabilities of being asleep per 30-minute interval across a day, for sleep/wake patterns averaged across 7-day windows (see Methods).

We first present results for the infant with the longest recording (541 days, Infant 1, Figs 1A and 2A). Fitting each daily sleep pattern independently showed that the model is able to capture the main features of the data (Fig 2B). Namely, the initial phase up to day $\sim 100$ where daily rhythms are emerging and sleep is highly fragmented, the onset of entrainment at a fixed phase at day $\sim 120$ where the consistent sleep pattern of two daily naps emerges, and persistence of this regime up to around day 280 where the sleep pattern evolves from two naps to a single nap each day. While the single nap sleep pattern is consistent throughout the second half of model fit data, the timings of waking and falling asleep do not completely align with the data. At the start of the trajectory (up to day $\sim 100$) there is large variability in $\chi$ and $v_{vc}$, with these parameters exhibiting compensatory effects (decreases in $\chi$ coincide with decreases in $v_{vc}$ and vice versa). In the first 100 days, the sleep patterns in the infant's data differ markedly from day to day, leading to the averaged sleep probability being relatively uniform across the day (Fig 2A).

The trajectories of each parameter in the best-fitting combinations (Fig 2C), although noisy, exhibit clear trends. For example, $\mu$ (the rise in homeostatic pressure from wake) is high in the early months of Infant 1, reaching a maximum of 13.8 nM s$^{-1}$ for the windows centred

on days 17 to 21, before settling around $\mu$ = 8 nM s$^{-1}$ for the rest of the trajectory (mean (SD) after timepoint centred on day 120 of $\mu$ = 7.97 (0.388)). The circadian strength parameter $v_{vc}$ starts close at zero (aligning with prior observation that there is negligible circadian influence on sleep/wake cycling at birth [24, 25]), before increasing in strength to between -2.25 to -4.5 mV after day 100. The homeostatic clearance time, $\chi$, starts small at around 5–10 h for some time (smoothed $\chi$ estimate below 10 h until timepoint centred on day 30), then increasing over time to eventually becoming relatively constant at between 30 and 35 h from day 130 to day 265, then increases again to 45 h from day 280 onwards. The parameter $b$ is high ($b$>0.95, versus adult $b$ = 0.4) and shows a slight decrease over time. While there are clear trends, there also exists evidence of non-monotonicity in how the parameters change with development.

The variability in TSD in Infant 1 is reflected in the variations in $\mu$ (Fig 2C). The sleep patterns in the early days of Infant 1 reflect the lack of discernible rhythm that is typical of newborns, which is also evident in the large variability in the parameters related to sleep timing ($\chi$, $v_{vc}$, $b$) in the early days (Fig 2C). Although the model captures the highly polyphasic nature of the sleep patterns and weak entrainment to clock time, this variability reflects the low identifiability of the parameters given the uncertainty in the empirical sleep probabilities. Many different parameter combinations can produce patterns that produce similar probabilities when averaged across a number of days. For example, for the earliest days in the infant's data (week centred on day 5 in Fig B in S1 Text) there is considerable density at the lowest end of the cost distribution across the grid, indicating that there are many parameter combinations that produce behaviours that have comparable similarity to the sleep patterns at the start of the data. Later in the data when there is a clearer sleep pattern, the lowest cost exists at the end of a thin tail, indicating increased identifiability of the model parameters with age.

## Trends of best-fitting parameter combinations for all infants

We repeated the fitting for the other infants. The trajectories of best-fitting parameter combinations for each infant are shown in Fig 3 (smoothed trajectories overlaid as an illustrative aid). We anticipated differences between the individual trajectories, as the sleep patterns were noticeably different between infants (Fig 1), for example with Infant 1 changing to a sleep pattern of 1 nap per day before 300 days of age and Infant 2 not changing to 1 nap per day until after 500 days of age. Nevertheless, there were many commonalities.

For Infants 1 and 3, we found an early rapid decrease in $\mu$, while Infant 2 starts much later (3 months) so appears to have missed that initial decrease. Infant 4 has a slower decrease in $\mu$, possibly due to the lack of parental intervention in sleep patterns [25] as the parents of Infant 4 explicitly did not influence the sleep patterns of the infant. Hence the slower decrease in $\mu$ in Infant 4 may be a more natural decrease, compared to the faster decreases with the other infants possibly being influenced by parental sleep schedule decisions. All four infants appear to have individual-specific best fit values of the parameters, with the differences in $\mu$ reflecting the individual differences in TSD.

Consistent among the datasets was the value of the parameter $b$, which predominantly controls the phase response of the circadian drive to the solar curve. After the initial newborn days, the best-fitting parameter combinations in each of the datasets tend to have values of $b \geq 0.9$, higher than the typical adult value of 0.4 [16, 29], indicating a phase-delayed response to the solar curve compared to adults. The Infant 1 trajectory yields $b$ values consistently 0.9–0.95 around the time consistent sleep bouts emerge, similar to Infant 2 which reaches $b$ = 0.95 at around day 300 when a consistent 2 nap sleep pattern emerges.

One consistent feature of the parameter trajectories is a positive relationship between $\chi$ and $v_{vc}$. While it was expected that $v_{vc}$ would become more negative (increase in strength [24, 25])

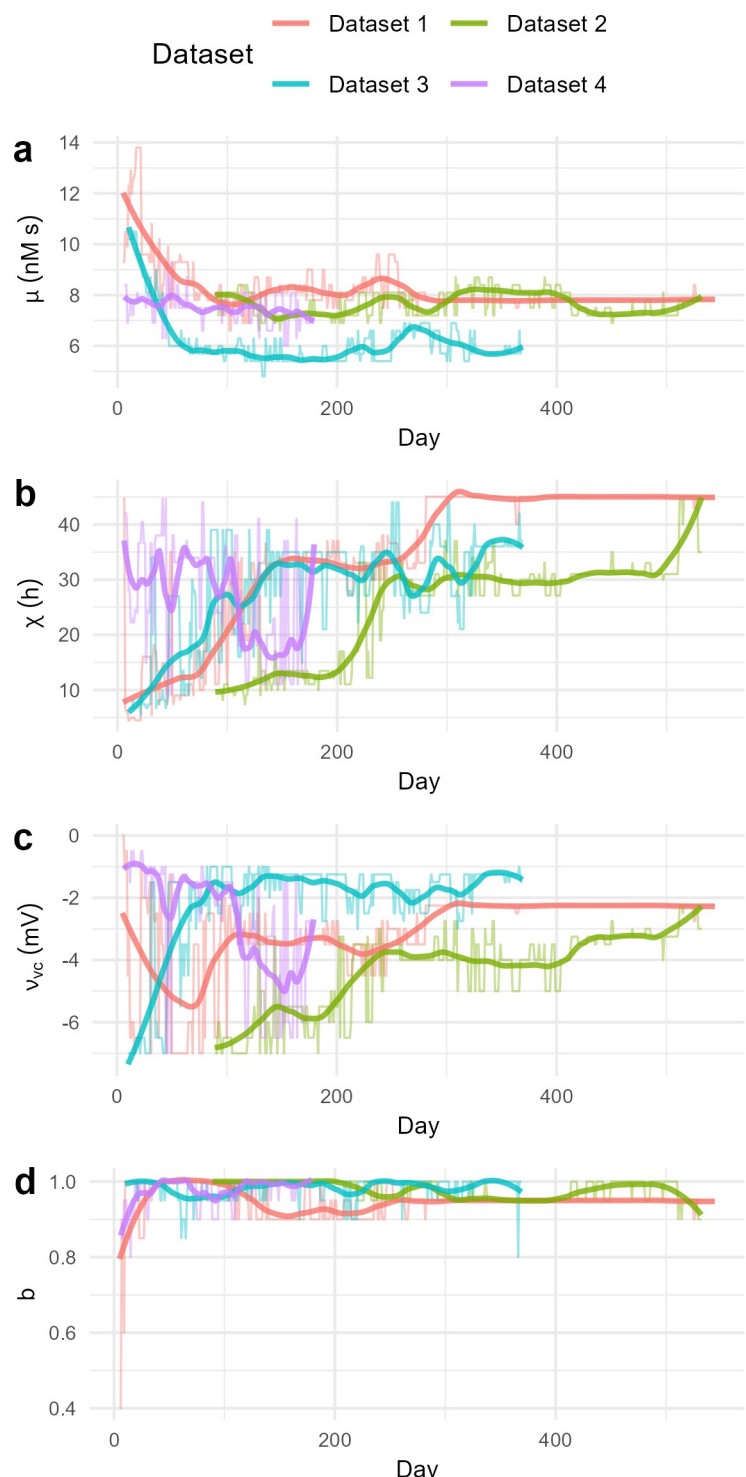

**Fig 3. The trajectories of best-fitting parameter combination for each infant.** For each parameter and infant, the best-fitting parameter value at each day (thin lines) is plotted with a Loess smoothed line (thick lines) with smoothing parameter $\alpha_{Loess}$ = 0.2. The day is the middle day of the sliding 7-day window.

with age, and $\chi$ would increase in age (to go from many bouts of sleep per day to consolidation [21]), the compensatory effects of these two parameters obscures the main trend in both parameters. In Datasets 1, 2 and 3, there is a rapid increase in $\chi$ before becoming mostly steady across time and only increasing when the number of bouts per day changes in the sleep patterns (around day 280 in Infant 1, and day 500 in Infant 2). Infants 1 and 4 both have rapid decreases in $v_{vc}$ during the youngest days, while Infants 2 and 3 start with strong $v_{vc}$ (large negative) before weakening, largely corresponding to changes in $\chi$. Infant 2 doesn't begin until 3 months of age, so it is possible that a rapid increase in the strength of $v_{vc}$ would have occurred before then.

The strong $v_{vc}$ values at the beginning of some of the trajectories (i.e., Infants 2 and 3) could reflect parental intervention in the timing of sleep bouts throughout the day, rather than signal from the SCN. Infant 4 (where the parents aimed to not explicitly influence the sleep patterns of the infant), began with a small $v_{vc}$ as the best fit before increasing in strength. Infant 4 also had a larger $\chi$ than Infant 1 and 3 in the first 100 days (though still less than the default adult value of 45 h $\approx$ ln 3.8). A $\chi$ value of $\approx$45 h does not occur in the trajectories of best-fit parameter combinations until after day $\sim$280 in Infant 1. This suggests that even with a range of possible $v_{vc}$ values, a lower $\chi$ is still needed during infancy and early childhood, at least until the sleep pattern is highly consolidated. It is possible that without parent intervention, Infants 1, 2 and 3 would have displayed sleep patterns that would have fit to smaller $|v_{vc}|$ and higher (though not as high as typical adult) $\chi$.

## Relative contributions of parameters at different ages

Finally, we asked whether reduced parameter spaces permit more parsimonious fits to the data, and whether different parameter combinations are more/less informative at different ages. We systematically fitted the model to Infant 1, allowing only constrained subsets of two or three parameters to vary, and assessed goodness of fit in terms of their cost (see Methods, Eq 14) relative to fitting all four parameters (Fig 4). For the four-parameter fit (black line in Fig 4A), the cost function value over time starts low, increases to a relatively stable value until a decrease just after day 300 where the one-nap regime is well fitted (Fig 2A and 2B), before increasing through to the end of the dataset where the model fails to capture the progressively later evening sleep onset (Fig 2A and 2B). Systematically fixing one parameter (cool colours in Fig 4A) or two parameters (warm colours in Fig 4A) naturally increases the fit cost, but some parameter combinations fit better than others. We found that allowing $\mu$ to vary from its adult value was essential, as fixing $\mu$ to an adult value ($\mu$ = 4.5 nM s) drastically increased the cost function value at all ages (dashed line in Fig 4A). For the remaining subsets with fixed $\mu$, we used $\mu$ = 8.1 nM s (grid value of $\mu$ that was the best fit for most of the time when Infants 1 and 2 had consolidated, consistent sleep patterns, cf. Fig 3). We also found that fixing $\chi$ to an adult value ($\chi$ = 45 h) tends to increase the fit costs, with those subsets tending to fit worse than those fixing $v_{vc}$ and $b$ ($v_{vc}$ = −3mV as most of best fit $v_{vc}$ values were between -2 and -4 mV so we selected the middle of that range, and $b$ = 1 given most of the infants best parameter fit involved $b$ = 1).

The different parameter subsets performed differently at different ages (Fig 4B and 4C). In the first 200 days of the dataset, fixing one parameter tended to not increase the cost significantly (increases of 8–37%) as compared to ages 12–18 months (last 200 days of Infant 1, increases of 38–91%). Fixing $\chi$ resulted in the highest increase in total cost, while fixing $\mu$ at a value consistent with infant and child sleep (8.1 nM s) gave the best overall three-parameter fit, albeit at some cost relative to the four-parameter fit indicating that $\mu$ is still informative over these ages. The increase in cost from fixing $v_{vc}$ to an adult value decreased with age, and the increase in cost from fixing $b$ to an infant value of 1.0 increased with age, supporting the

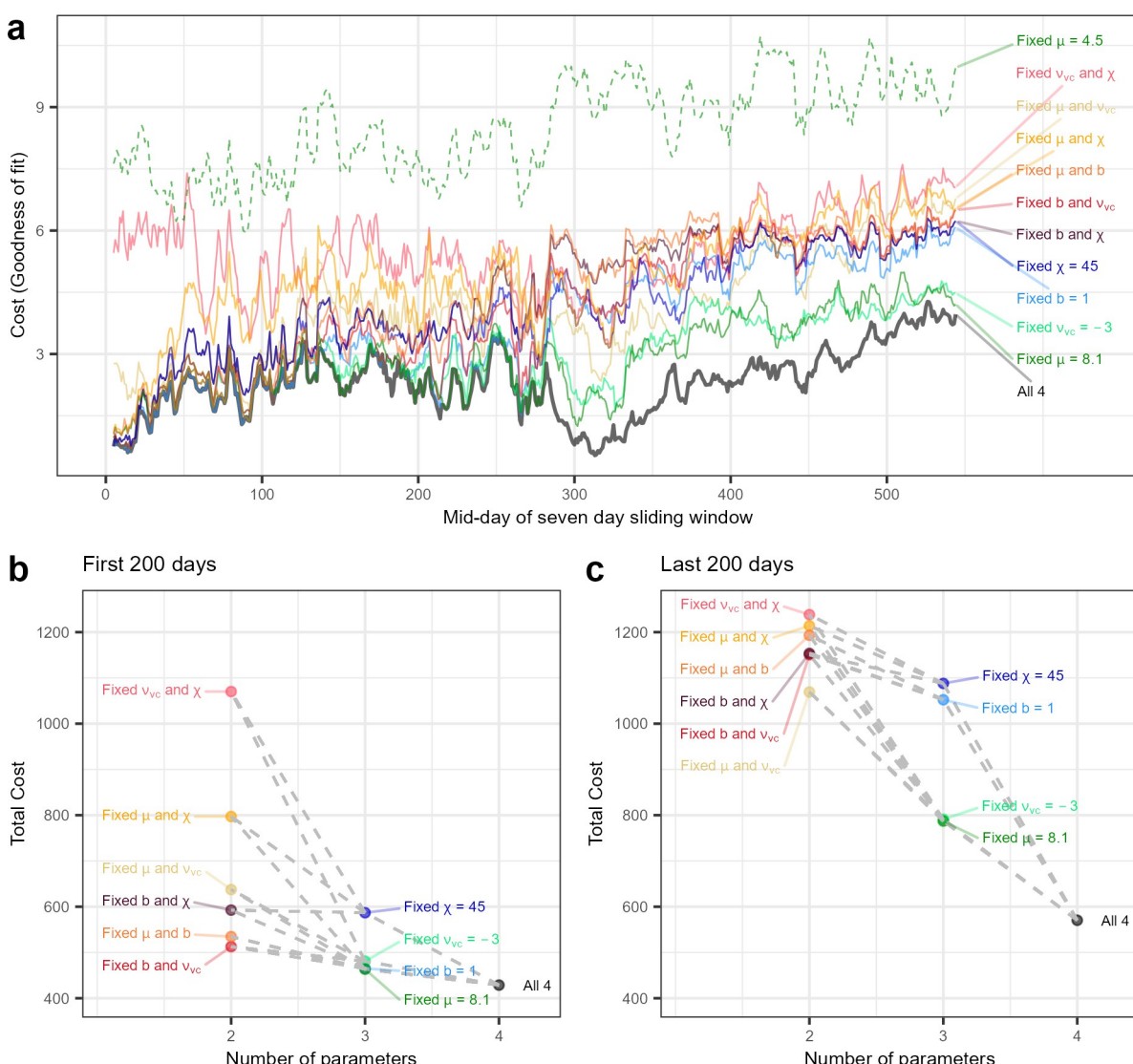

**Fig 4. Relative contributions of subsets of parameters at different ages in infancy. *a)*** The minimum cost function value over time for different parameter constraints in Infant 1. ***b)*** The total cost value over the first 200 days of Infant 1 (seven day windows from midday 5 to 204). ***c)*** The total cost value over the last 200 days of Infant 1 (seven day windows from midday 345 to 544). Subsets of parameters with two fixed parameters have the fixed value from the corresponding single fixed parameter sets (connected by dashed lines in b and c), that is $\mu = 8.1$ nM s$^{-1}$, $\chi = 45$ h, $v_{vc} = -3$ mV, and $b = 1$.

notion that the circadian rhythm matures over infancy. At young ages, the best two-parameter fits perform similarly to three- and four-parameter fits (Fig 4A and 4B) and the distribution of costs has a higher density towards the minimum (Fig B in S1 Text). The three lowest cost two-parameter fits each fix the parameter *b*, suggesting *b* = 1 could be a parsimonious solution for the first three months of age, and the two highest cost two-parameter fits both fix $\chi$, indicating the importance of varying $\chi$ in early infancy (Fig 4B).

## Discussion

Computational modelling of sleep has focused on adult sleep, leaving the rapid and profound development of sleep patterns across infancy and early childhood largely unexplained. Here,

we used a computational model to infer the physiological changes that drive normal sleep maturation. The model reproduced the observed changes in the probability of being asleep in 30-minute time bins throughout the day in the first 2 years of life, reflecting the characteristic maturation of sleep duration, sleep fragmentation, and sleep timing. We found that the underlying biological processes appear to mature at different rates, with the strength of circadian rhythms increasing rapidly after birth, sleep homeostatic accumulation and the circadian system's response to light maturing slowly, with these processes likely continuing into childhood or adolescence, and sleep homeostatic clearance rapidly slowing over infancy.

Our findings reveal an important role for four key parameters in the maturation of sleep/wake patterns. Two of these parameters reflect changes in the sleep homeostatic process, one represents the strength of the circadian drive, and one represents the relative sensitivity of the circadian clock to phase-delaying light. While there were similar trends between the four infants we examined, there were also notable differences in when and how the trajectories evolved. This is consistent with the large heterogeneity in infant sleep behaviour [3, 5]. In adults, it is well established that there are large inter-individual differences in sleep and circadian physiology [35–38]. While inter-individual differences in sleep and circadian physiology are not yet well understood in infants, our work provides a novel method for potentially inferring these differences. This approach could be used to understand the reasons for pathological sleep conditions and to propose solutions tailored to an individual's physiology.

In previous work, this computational model has been used to simulate sleep/wake patterns in older age groups, from adolescents through to older adults [23]. However, this is the first time the model has been applied to simulating sleep/wake patterns in infants. Compared with previous fitting of the same model to adults, our results reveal systematic differences in inferred physiological parameters, which in most cases align with theories of circadian and sleep homeostatic development. Specifically, we found shorter timescales for sleep homeostatic clearance and accumulation. These findings are consistent with previously proposed theories [27] and consistent with previous modelling work showing that such homeostatic differences drive more rapid sleep/wake cycling in children [39], and in species that sleep polyphasically, similar to human infants [21,22]. This result also aligns with that of Jenni, Achermann and Carskadon [40] and Skeldon, Derks and Dijk [23], whereby the rate of accumulation of sleep pressure during wakefulness ($\mu$) continues to decrease throughout adolescence from the higher childhood values to the lower adult value. We also found circadian amplitude was generally weaker in the first ~100 days of life compared to adults, consistent with the observation that it takes time for robust diurnal rhythms to emerge after birth [24,25] and evidence from animal studies that the circadian system is still undergoing development after birth [32]. Finally, we inferred an increased sensitivity to phase-delaying light in infants compared to adults, consistent with a recent study showing quite large phase delays in children in response to evening light stimuli [41].

An interesting observation from our findings was that the enhanced sensitivity to phase-delaying light remained robustly the case across the full age range that we explored. We inferred $b \approx 1$ in each of the four infants' data versus $b \approx 0.4$ in adults. This suggests that maturation of the circadian phase response to light may occur over a longer time period. This seemingly aligns with findings showing enhanced sensitivity of the circadian system to light in young children based on melatonin suppression to evening light [42], and greater light sensitivity in early-to-mid adolescence compared to late adolescence [43]. Collectively, these findings suggest that there are likely pubertal changes in how the human circadian system responds to light.

Newborn and early infant sleep is characterized by high BPD, with many night awakenings and bouts of sleep throughout the day. Both fast homeostatic clearance (low $\chi$) and weak

circadian drive (low $|v_{vc}|$) can produce polyphasic sleep (BPD > 1), though to produce BPD > 4 a lower $\chi$ is required. While it is expected that the circadian drive to sleep would start at close to 0 before strengthening [24,25], our modelling suggests that both $\chi$ and $v_{vc}$ have different values in infancy versus adulthood, and thus developmental changes in both parameters would occur during infancy. Given the complementary relationship between the two parameters (Fig A, panel c in S1 Text), where changes in one can be offset by changes in the other to have consistent BPD, it is difficult to pinpoint exactly how each parameter would change without a further constraint. Such a constraint could be informed by animal studies into the development of the sleep homeostatic process in infancy (e.g. sleep deprivation studies), or modelling circadian markers such as cortisol or core body temperature.

There are some important limitations to this work. First, we made several simplifying assumptions in this model, which could be investigated as ways to refine model performance in future work. For instance, the relatively simple assumptions regarding the environmental light exposure patterns and the model's freedom to sleep and wake freely. We have not modelled other factors that could affect sleep. For example, in infants the timing of each sleep bout can be influenced by feeding needs, the external environment (e.g. loud noises), and an inability to self soothe after brief awakenings. Moreover, interventions by parents/carers can change an infant's sleep schedule, for example by encouraging naps at certain times, attempting to consolidate naps, or promoting sleep onset at night. While we expect development is a continuous process, there exists in the model parameter space between the 2-nap and 1-nap sleep behaviour, combinations of parameters that produce behaviour which alternates between 1-nap and 2-naps. Parents may intervene at this point to impose a 1-nap solution. These external forces may result in changes in parameters that do not actually reflect changes in the underlying physiological processes, thus reducing the model's accuracy and affecting the interpretation. Notably, this model can be used to simulate constraints on sleep timing [16,17,44], meaning that such behaviours could theoretically be incorporated to more accurately model individual circumstances. Future work could also incorporate effects of other environmental stimuli, such as noise [20] or sleep-encouraging behaviours from parents, to more accurately describe the inputs to the sleep model. While we have attempted to ameliorate stochastic effects by fitting to sleep patterns averaged over a week, influences that are consistent from day to day would affect our parameter estimates. Hence some sudden jumps in inferred parameter values may reflect the model attempting to compensate for external influences. The observed non-monotonicity in the trends may also be a result of these external forces. Other causes may be identifiability issues, as well as the interdependence between parameters leading to any non-monotonicity in one parameter being reflected in other parameters. Nevertheless, we expect that the longer and slower changes do reflect the main features of sleep maturation. We anticipate that incorporating noise in the model would lead to greater day-to-day variability in the sleep/wake patterns, and potentially a greater number of sleep bouts. This may change some of the parameter estimates.

Second, while we found differences in the parameter trajectories between the infants, the limited sample size makes it difficult to draw conclusions about the heterogeneity in sleep patterns in infancy, as well as the differences in recording methods (sleep diary vs actigraphy) which can bias estimates in TSD and sleep onset time [45] with the particular complications of using actigraphy in infancy well documented [46] including underestimating day time sleep and dependence on classification algorithm. Heterogeneity in recording methods likely reduces the identifiability of parameters at the cohort level; future work would benefit from large datasets with fixed recording protocols. Moreover, while Infant 4 [25] is an example of parents having as little input as possible to the sleep times, sleep timings from the other infants are influenced in some way by parent/carer decisions unique to each family. Third, while we

have selected physiological parameters in the model that are well motivated by theory and by empirical evidence, we cannot exclude the role of other physiological mechanisms in also contributing to developmental changes in sleep/wake patterns.

While previous sleep/wake modelling has covered the period from early adolescence to old age, the work presented here extends our understanding of sleep/wake regulation into infancy. We present proof-of-principle for individualized estimation of parameters governing the development of human infant sleep patterns. It should now be possible to map the full lifespan trajectory in sleep maturation as detailed data become available, which is increasingly feasible with the rise of sleep diary apps and wearables. Beyond understanding natural development, our modelling framework could also be used to understand the underpinnings of sleep disorders in children, for example by inferring the model parameters for specific disorders, e.g. sleep disordered breathing [47]. This could lead to an improved understanding of the underlying neurobiology of such disorders, as well as revealing model informed ways to address these issues. The model could also be used to develop optimised sleep schedules for children, where improvements in sleep could lead to improvements in neurodevelopment and cognition, similar to work that has been done for adolescence [16,48]. We thus anticipate that sleep modelling will be a valuable tool in paediatric sleep research.

## Methods

### Sleep/wake switch model

The mutual inhibition between the MA and VLPO neuron populations (red and blue respectively in Fig 5) is governed by Eqs 1 and 2 [19], respectively,

$$\tau_m \frac{dV_m}{dt} = -V_m + v_{mv}Q_v + A, \tag{1}$$

$$\tau_v \frac{dV_v}{dt} = -V_v + v_{vm}Q_m + D, \tag{2}$$

where

$$Q_j = \frac{Q_{\max}}{1 + \exp(-(V_j - \theta)/\sigma)}, j = m, v. \tag{3}$$

The mean firing rate and mean cell body potential of the neuron populations $j = m,v$ are $Q_j$ and $V_j$, respectively, and $A$ is excitatory input from the orexinergic and cholinergic neurons [19,49]. The $v_{ab}$ terms represent the strength of the input/connection from the neuron population/drive $b$ into neuron population $a$. For example, $v_{mv}$ is the strength of the VLPO input into the MA group.

The sleep drive to the VLPO is the term $D$ in Eq 4, which is the weighted sum of the sleep homeostatic process ($H$), circadian process ($C$), and constant background input to the VLPO ($D_0$) which represents the sum of all other inhibitions and excitations to the VLPO [20]:

$$D = v_{vh}H + v_{vc}C + D_0. \tag{4}$$

The sleep homeostatic process $H$, the process by which sleep-inducing chemicals such as adenosine build up in the brain (particularly the basal forebrain) during wakefulness and clear during sleep, obeys

$$\chi \frac{dH}{dt} = -H + \mu Q_m, \tag{5}$$

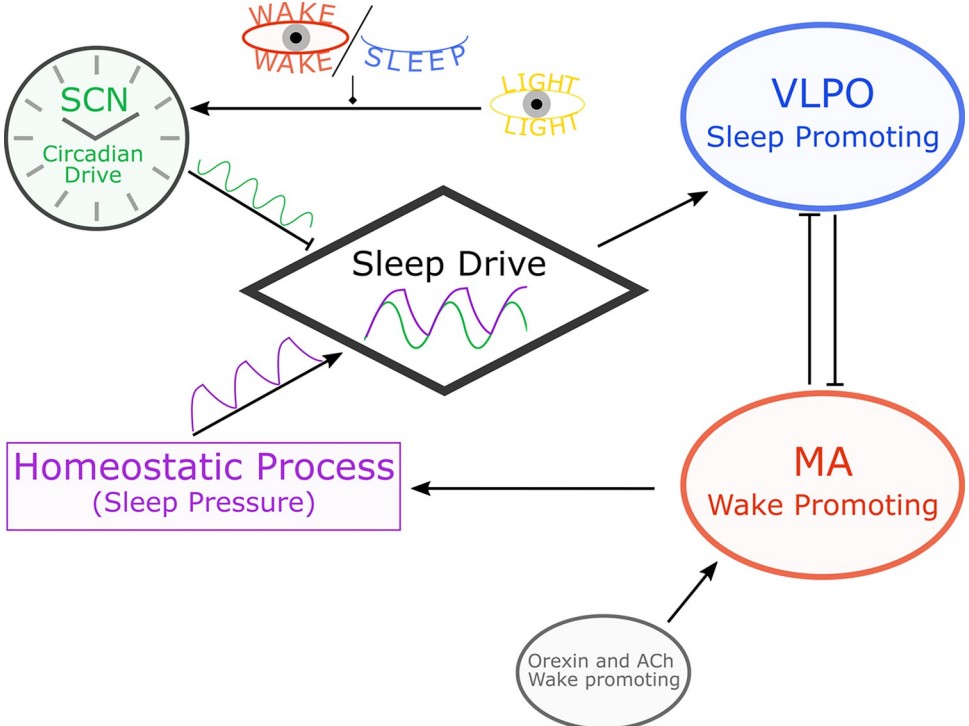

**Fig 5. Schematic of the sleep model.** The sleep/wake states are produced by mutual inhibition between the sleep-promoting VLPO (blue) and wake-promoting MA neuronal populations (red). The MA receives excitatory input from orexin neurons and cholinergic (Ach) neurons. The sleep drive is a combination of a homeostatic sleep drive (purple) and a light-entrained circadian drive (green). The circadian drive is entrained by the daily light cycle (yellow), and gated by the arousal state (no influence from light when asleep). Arrows denote excitatory connections, flat line ends denote inhibitory connections.

where $\chi$ is the characteristic clearance time of the somnogens, and $\mu$ is the rate of increase in the level of somnogens, assumed to be proportional to the activity of the wake-promoting group MA, due to the correspondence between MA activity and arousal state [19].

The circadian process $C$ is defined by

$$C(t) = \frac{1}{2}(1 + 0.80y - 0.47x), \tag{6}$$

where $x$ and $y$ are the variables of a forced, modified van der Pol oscillator [28] defined by

$$\kappa\frac{dx}{dt} = \gamma\left(x - \frac{4x^3}{3}\right) - y\left(\left(\frac{24}{f\tau_c}\right)^2 + kB\right), \tag{7}$$

$$\kappa\frac{dy}{dt} = x + B, \tag{8}$$

$$\frac{dn}{dt} = \lambda\left(\alpha_0\left(\frac{\tilde{I}}{I_0}\right)^p(1-n) - \beta n\right), \tag{9}$$

where $n$ is the fraction of activated photoreceptors and $B$ is the drive from the photoreceptive

**Table 1. Parameter values used in modelling.**

| Sleep/wake regulation | | | | Circadian | | | | Light | |
|---|---|---|---|---|---|---|---|---|---|
| Parameter | Value | Parameter | Value | Parameter | Value | Parameter | Value | Parameter | Value |
| $Q_{max}$ | 100 s$^{-1}$ | $\nu_{vh}$ | 1.0 mv nM$^{-1}$ | $\alpha_0$ | 0.16 | $\kappa$ | 12/$\pi$ h | $c$ | 1/6000 |
| $\theta$ | 10 mV | $\nu_{vc}$ | * mV | $I_0$ | 9500 lux | $\gamma$ | 0.23 | $l_e$ | 20 lux |
| $\sigma$ | 3 mV | $D_0$ | -10.2 mV | $p$ | 0.6 | $f$ | 0.99669 | $l_d$ | 10000 lux |
| $A$ | 1.3 mV | $\chi$ | * h | $b$ | * | $\tau_c$ | 24.2 h | $s_1$ | 8 h |
| $\nu_{vm}$ | −2.1 mV s | $\mu$ | * nM s | $G$ | 19.9 | $k$ | 0.55 | $s_2$ | 17 h |
| $\nu_{mv}$ | −1.8 mV s | | | $\lambda$ | 60 | | | | |
| $\tau_{m,j}$ | 10 s | | | $\beta$ | 0.013 | | | | |

*no fixed value as it is part of the grid of values

pathway to the circadian pacemaker [50],

$$B = G(1 - n)(1 - bx)(1 - by)\alpha_0 \left(\frac{\tilde{I}}{I_0}\right)^p, \tag{10}$$

$$\tilde{I} = \begin{cases} L(t), & V_m > V_v \\ 0, & V_m \leq V_v \end{cases}, \tag{11}$$

and where $L(t)$ defines the light level, with

$$L(t) = l_e + \frac{(l_d - l_e)}{2} \tanh(c(t - s_1)) - \tanh(s(t - s_2)). \tag{12}$$

This light function describes a combination of brighter exposure to light levels during the day (due to a combination of indoor and outdoor light sources), and exposure to indoor artificial light during other hours of wakefulness. The minimum light level in the evening is given by $l_e$, and maximum light level during the day is given by $l_d$. The term $c$ defines the steepness of the solar curve, with the change from $l_e$ to $l_d$ occurring around the time $s_1$ (dawn), and the change from $l_d$ to $l_e$ occurring around the time $s_2$ (dusk). Values for $s_1$, and $s_2$ were taken from Skeldon, Phillips and Dijk [16]. We used 20 lux for $l_e$ and 1000 lux for $l_d$ to more closely match the light infants and young children would experience while awake [51].

Wake is defined as when $V_m > V_v$. Parameter values are given in Table 1.

## Implementation

We implemented the model in MATLAB R2021a using the ode23s solver. The model was solved for all combinations of parameter values as listed in Table 2, for an initial time of 4 weeks to remove transients caused by initial values, then using the last value of the initial run as the starting values, solved for a time period of 35 days to produce a sleep/wake state time series. We calculated sleep characteristic summary measures on the last 14 days of the generated time series. To create the windows of sleep probability (e.g. Fig 2B), we interpolated the time series to have uniform time points, and the probability of being asleep in each half an hour time window was calculated on the last seven days of the time series.

## Sleep characteristic summary measures

There is considerable variability in how studies characterize sleep at different ages. Here we focus on two measures that are applicable across the age range explored: the total sleep

**Table 2. Parameter ranges defining the grid search parameter space.**

| Parameter | Description | Adult value | Grid Search Range |
|---|---|---|---|
| $\chi$ | Characteristic clearance time for sleep homeostatic pressure. Lower value results in sleep pressure dissipating faster. | 45 h (ln(45) = 3.8) | 15 h to 45 h in steps of 1 h, and for ln $\chi$, 1.5 to 3.8 in steps of 0.1 (ln $\chi \in [1.5, 3.8], \chi \in [4.5\,h, 45\,h]$) |
| $\mu$ | Accumulation rate for sleep homeostatic pressure during wake. Higher number means faster accumulation. | 4.2 nM s | 2.1 nM s to 15.9 nM s in steps of 0.3 nM s |
| $\nu_{vc}$ | Strength of circadian drive. More negative values mean a stronger circadian drive. | -3.37 mV | -7 mV to 0 mV in steps of 0.5 mV with additional steps from -3.5 mV to -1.0 mV in steps of 0.25 mV |
| $b$ | Parameter determining shape of the circadian clock's phase-response curve to light. Higher values bias the phase-response curve toward phase delay. | 0.4 | 0.4 to 0.7 in steps of 0.1, 0.8 to 1.0 in steps of 0.05 |

duration (TSD) per 24 h (midnight to midnight), and the number of bouts of sleep per 24 h (BPD, in particular in the model, the number of state transitions per 24 h divided by 2).

## Parameter grid search

We performed simulations varying four parameters that could plausibly vary with age and which have been previously shown to affect sleep patterns in a way that could account for developmental changes. The specific parameters used and their ranges are presented in Table 2. The circadian coefficient $\nu_{vc}$ is negative, as the circadian rhythm acts to inhibit sleep in the model. Because of the nonlinear relationship between $\chi$ and the number of sleep bouts per 24 hours, we initially varied $\chi$ on a logarithmic scale. A further set of $\chi$ and $\nu_{vc}$ values were added to improve fitting given the effect of $\chi$ on sleep timing [28]. We set physiological bounds on each parameter based on previous parameter range estimates and biological constraints [29,49]. The ranges of $\chi$ and $\mu$ were chosen to be able to produce the infant sleep characteristics of large total sleep durations (high $\mu$) and many bouts of sleep (low $\chi$). We updated the feasible constraints of $\mu$ to allow infant sleep, as per Phillips and Robinson [49], using an upper bound of infant sleep and wake times of 18 and 6 hours [3], the explored range of $\mu$ was increased to 15.9 nM s.

## Extracting and converting the empirical data

Infant 1:

Sleep diary data taken from the Agenoria package (https://github.com/jiuguangw/ Agenoria) collected from a male infant by Jiuguang Wang (www.robo.guru), used for academic purposes under CC BY-NC-SA 4.0. Data was recorded as sleep onset and offset time. We used only complete days of data.

Infant 2:

Sleep diary data taken from the Baby-data-viz package (jitney86 https://github.com/ jitney86/Baby-data-viz), used for academic purposes under MIT license. Data was recorded as whether the infant was awake, awake and feeding, or asleep, in 15 minute intervals. We used only complete days of data. No information on biological sex given.

Infant 3:

Actigraphy data from a female infant, converted to rest/activity using a cut-off of 5 activity counts per minute as per Jenni, Deboer and Achermann [24]. Periods of one hour or more with zero activity were considered missing data and removed.

Infant 4:

Sleep diary data taken from a scan of the published image from McGraw, Hoffmann, Harker and Herman [25]. Data from a male infant was originally manually recorded, and was

published with feed times, sunsets, sunrises and other information overlaid. The image was scanned in black and white, imported to Matlab as numeric array via the imread function (scale from 0 to 255 where black is 0 and white is 255), and a cut-off used to distinguish between wake and sleep (cut-off of 200). Horizontal lines of pixels were manually chosen for extraction to represent each day. Extra information (sunrise, sunsets etc.) was removed manually.

It should be noted that the data collection method for each subject is different, which is a clear limitation to this work.

### Fitting to empirical data

To facilitate fitting, for both the data and the model simulations we calculated the probability of being asleep by clock time, using half-hour clock time bins, computed using 7-day sliding windows for averaging (long enough to aggregate data, and shorter than the timescale on which most developmental changes occur) with 6-day overlap (sliding step of 1 day). We assessed similarity between the model and the data using a cost function based on sum of squares:

$$Cost = \sum_{30 \; minute \; windows} \left(prob_{data} - prob_{grid}\right)^2, \tag{13}$$

With Total Cost normalised to a time period of $n$ days given by,

$$\text{Total Cost} = \frac{1}{n} \sum_{n \; days} \text{Cost}, \tag{14}$$

The parameter combination that minimized the cost function was identified for each sliding window week in the sleep datasets.

Since the sleep/wake patterns in each of the empirical datasets were not fully consolidated (all included daytime napping), we used only parameter combinations that produced times series with an average bouts per day of 1.5 or more. This allowed the trajectory algorithm to only consider non-consolidated sleep/wake patterns, while not biasing the fitting toward time series with a pre-specified number of bouts. We also limited the search space for the best-fitting parameter combinations to those with a pacemaker period close to 24 hours (between 23.995 h and 24.005 h). Ties in optimal cost function value between parameter combinations (which could often occur when $v_{vc} = 0$ resulting in $b$ having no effect on sleep patterns) were decided based on the closest pacemaker period to 24 h. The period ($\tau_{obs}$) of the circadian pacemaker (the modified forced van der Pol oscillator, Eqs 7 and 8) was calculated by from the oscillator's rate of phase accumulation:

$$\psi = \arctan(y, x), \tag{15}$$

$$\tau_{obs} = \frac{2\pi}{(\psi_{end} - \psi_{start})/(t_{end} - t_{start})}. \tag{16}$$

### Supporting information

**S1 Text. Supplementary 1 –Selection of results from grid sweep.** Supplementary 2 –Distribution of cost function value for Infant 1. Supplementary 3 –Comparison of optimal parameter combinations with and without period constraint. Supplementary 4 –Summary of fitted parameters for specific sleep pattern periods.
(PDF)

## Acknowledgments

We thank Oskar Jenni and Kate McGraw for assistance with Infants 3 and 4.

## Author Contributions

**Conceptualization:** James A. Roberts.

**Data curation:** Lachlan Webb.

**Formal analysis:** Lachlan Webb.

**Methodology:** Lachlan Webb, Andrew J. K. Phillips, James A. Roberts.

**Project administration:** Lachlan Webb.

**Supervision:** James A. Roberts.

**Visualization:** Lachlan Webb.

**Writing – original draft:** Lachlan Webb.

**Writing – review & editing:** Lachlan Webb, Andrew J. K. Phillips, James A. Roberts.

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
