## [Decision Letter · Decision Letter 0]

18 Apr 2024

Dear Mr Webb,

Thank you very much for submitting your manuscript "Mapping the physiological changes in sleep regulation across infancy and young childhood" for consideration at PLOS Computational Biology.

As with all papers reviewed by the journal, your manuscript was reviewed by members of the editorial board and by several independent reviewers. In light of the reviews (below this email), we would like to invite the resubmission of a significantly-revised version that takes into account the reviewers' comments.

We cannot make any decision about publication until we have seen the revised manuscript and your response to the reviewers' comments. Your revised manuscript is also likely to be sent to reviewers for further evaluation.

Sincerely,

Leila Tarokh

Guest Editor

PLOS Computational Biology

Thomas Serre

Section Editor

PLOS Computational Biology

Reviewer's Responses to Questions

**Comments to the Authors:**

Reviewer #1: The manuscript presents a mathematical model attempting to explain developmental changes in infant sleep. Applying this model to longitudinal sleep data from 4 infants confirms the faster accumulation and clearance of sleep homeostatic pressure and the weaker circadian rhythm in this sample as compared to previous observations in adults. Additionally, greater sensitivity to phase-delaying effects of light is observed in the presented dataset. Taken together, this work represents an important step to understanding the mechanisms contributing to the large inter-individual variability in sleep regulation during infancy. The integration of knowledge from different fields including neurobiology and computational modelling to address this complex question is a key strength of the study. However, the limitation stemming from the fact that the four datasets correspond to only four individual subjects should be more explicitly acknowledged in the manuscript, as it dampens overall enthusiasm for the study's findings. Furthermore, the manuscript would benefit from a more nuanced discussion of the contextual factors influencing infant sleep as well as future directions regarding mathematical modeling thereof. Finally, I recommend several minor changes to improve the readability of the manuscript (see below).

Results

Sleep-wake switch model (page 3): I suggest moving Figure 1 to the Methods section in order to make clear that this is a previously validated model that was used and not developed in the scope of the current study.

Empirical sleep patterns (page 3): I recommend referring to e.g. “subjects” instead of “datasets” throughout the manuscript to make clear that data from four individuals was used in the study.

The parameter space of sleep maturation (page 5, “…the circadian rhythm in sleep is non-existent in the days soon after birth.”): I strongly recommend rephrasing this sentence to accurately reflect the body of evidence indicating that circadian rhythms begin developing already before birth. The term 'non-existent' is overly strong; although sleep patterns in the first few weeks may not yet resemble those of adults or older children, there is significant variability among infants. In some cases, clear diurnal tendencies can be observed shortly after birth.

Trajectories through parameter space inferred from empirical data (page 5-6): I believe this section would benefit from a more precise and quantifiable presentation of the results, rather than the current descriptive approach. For example, in the last sentence on page 5, the authors mention that the homeostatic clearance time “starts small at around 5-10 h for some time”, which is very vague.

Trends of best-fitting parameter combinations for all datasets (page 7): I appreciate that parental interventions are acknowledged as a potential confounding factor. However, I believe that including further information about what these parental interventions included, and under which circumstances the data were collected is important for the interpretation of the current findings. I understand that this information is provided elsewhere, however, given its relevance to the current study, I believe it should be directly accessible without the need to consult other publications.

Relative contributions of parameters at different ages (page 9): I believe the figure reference should read “Figure 5a” instead of “Figure 6a”.

Discussion

Page 10 (“…sleep homeostatic clearance rapidly slowing, and sleep homeostatic accumulation…”): I believe sleep homeostatic clearance and accumulation should be reversed in the last sentence, particularly given that the findings regarding clearance are not as conclusive.

Page 11: In the last paragraph before the limitations, the sentence “…it is

possible that both developmental changes in both…” is not clear. I suggest rephrasing. The same paragraph would benefit from a more direct discussion of the implications regarding the relationship between the two parameters. Could the authors suggest alternative modeling approaches to overcome this issue? What would these “further constraints” encompass?

Page 12: The manuscript mentions the role of parental interventions and external environmental factors but does not provide a detailed discussion of how these might affect the model's accuracy or the interpretation of results. I recommend incorporating a more explicit elaboration on the implications of such factors for the current study’s findings as well as for similar modeling approaches in the future. Similarly, while the last paragraph notes the clinical relevance of such modeling approaches, the description of potential future directions remains superficial. I believe the conclusion would benefit from more specific suggestions for future applications.

Methods

Overall, I believe this section could be structured more clearly. For example, on several occasions, the variables used in the equations are introduced rather late. For example, the reader must wait until equation 4 to understand v which already appears in equation 1.

Furthermore, referencing previous work, particularly on page 13 would help the reader understand how established which parts of the modeling process are and how exactly the current study builds on this work to adapt the approach to the new application.

Extracting and converting the empirical data (page 16): I strongly recommend describing the four datasets in more detail. For instance, I believe the sex of the four infants is not mentioned in the manuscript. Additionally, the fact that the four datasets employ different methods for sleep data collection (sleep diary vs. actigraphy) should be acknowledged as a limitation, and their comparability should be further discussed. Furthermore, the difficulty of deriving reliable sleep data from infant actigraphy should be acknowledged (e.g., Schoch et al., 2019, SLEEP). Lastly, the methodology for deriving sleep data from dataset 4 is unclear. Could the authors provide more information, e.g., about the specific cut-off mentioned?

Fitting to Empirical Data (page 16): The 23.995 to 24.005-h period in the last sentence seems narrow. Can the authors justify this choice and have they tested other ranges?

Reviewer #2: uploaded as an attachment

**Have the authors made all data and (if applicable) computational code underlying the findings in their manuscript fully available?**

Reviewer #1: Yes

Reviewer #2: Yes

PLOS authors have the option to publish the peer review history of their article (what does this mean?). If published, this will include your full peer review and any attached files.

Reviewer #1: No

Reviewer #2: No
---

## [Decision Letter · Decision Letter 1]

25 Sep 2024

Dear Dr Webb,

Thank you very much for submitting your manuscript "Mapping the physiological changes in sleep regulation across infancy and young childhood" for consideration at PLOS Computational Biology. As with all papers reviewed by the journal, your manuscript was reviewed by members of the editorial board and by several independent reviewers. The reviewers appreciated the attention to an important topic. Based on the reviews, we are likely to accept this manuscript for publication, providing that you modify the manuscript according to the review recommendations.

Sincerely,

Leila Tarokh

Guest Editor

PLOS Computational Biology

Thomas Serre

Section Editor

PLOS Computational Biology

Reviewer's Responses to Questions

**Comments to the Authors:**

Reviewer #1: I thank the authors for addressing my comments. I believe the manuscript is suitable for publication in its current form and have no further concerns.

Reviewer #2: Overall, I’m ok. with the changes and clarifications made. However, one point still needs further clarification, as noted in the comment below.

It's unfortunate that no simulations were presented. While the paper focuses on parameter estimation and interpreting their changes over time, it has not been shown whether the estimated parameters lead to realistic sleep-wake patterns in simulations. Only the probability of being awake or asleep was demonstrated. Therefore, the statement in the discussion, “The model reproduced the characteristic changes in sleep duration, sleep fragmentation, and sleep timing in the first 2 years of life,” is an overstatement and should be removed or revised.

Minor: Supplementary Figure 3 a,b: Please use the same dimensions for plotting as in Figure 2 a,b to facilitate visual comparisons.

**Have the authors made all data and (if applicable) computational code underlying the findings in their manuscript fully available?**

Reviewer #1: Yes

Reviewer #2: Yes

PLOS authors have the option to publish the peer review history of their article (what does this mean?). If published, this will include your full peer review and any attached files.

Reviewer #1: No

Reviewer #2: No

Figure Files:

Data Requirements:

Reproducibility:

References:

---

## [Editor Report · Decision Letter 2]

7 Oct 2024

Dear Dr Webb,

We are pleased to inform you that your manuscript 'Mapping the physiological changes in sleep regulation across infancy and young childhood' has been provisionally accepted for publication in PLOS Computational Biology.

Best regards,

Leila Tarokh

Guest Editor

PLOS Computational Biology

Thomas Serre

Section Editor

PLOS Computational Biology

---

## [Editor Report · Acceptance letter]

15 Oct 2024

PCOMPBIOL-D-24-00193R2 

Mapping the physiological changes in sleep regulation across infancy and young childhood

Dear Dr Webb,

I am pleased to inform you that your manuscript has been formally accepted for publication in PLOS Computational Biology. Your manuscript is now with our production department and you will be notified of the publication date in due course.

With kind regards,

Dorothy Lannert
